# Take Another Look: Improving Information Extraction From Images With Multiple Encoders

## Abstract

Unstructured image data are increasingly used across diverse applications, yet standard practices for extracting predictive features remain underexplored. We redefine encoder choice as an ensemble and present three strategies for leveraging multiple pre-trained encoders to mitigate model risk associated with using a single encoder. These strategies include: feature union, which concatenates encoder features before model training, and two forms of model averaging, which weight predictions from single encoders using either equal weights or weights chosen with regression. Across six prediction applications—house prices from exterior images, poverty rates from satellite imagery, breast cancer and pneumonia detection from chest X-rays, rice disease classification from leaf images, and facial age —our results show three key findings: (i) using multiple encoders consistently outperform single encoders, with out-of-sample $R^2$ for house prices, for example, increasing from 15.01% (best single encoder) to 24.1% with feature union; (ii) model averaging reduces error rates across all classification tasks, for example from 13.31% to 8.68% offering consistent gains over both single encoders and feature union; and (iii) using multiple encoders methods mitigate model risk, as accuracy varies widely across individual encoders in different applications. These results demonstrate that using multiple encoders provides robust, high-performing pipelines for image-based prediction without requiring extensive task-specific fine-tuning.

## 1 Introduction

Advances in deep learning models have made it more common to incorporate images in research across many disciplines. Typically, this involves passing an image through a pre-trained encoder and using the resulting features for prediction tasks. Practitioners frequently default to using a single encoder, such as ResNet50. When more than one encoder is used, they are typically only used treated as rivals (e.g. Shah et al., 2023; Gaviria Rojas et al., 2022; Fu et al., 2022; Chen et al., 2021; Theckedath & Sedamkar, 2020).

Each convolutional neural network (CNN) architecture that makes up these encoders identifies different features for a given image—even if they are trained on the same images. However, these differences raise two important questions. First, does relying on a single encoder expose researchers to the risk that a particular encoder is not well-suited to a given task? And second, can combining encoders lead to improvements in predictive accuracy?

We answer these questions by using multiple encoders simultaneously to demonstrate how the choice of encoder limits the prediction accuracy in applications that have previously focused on single-encoder approaches. We introduce this in two ways. First, following Nordstrom et al. (2024), we use the features from all encoders ("feature union") to use them jointly in predictions. Second, we propose a model averaging approach that combines predictions using either equal weights across the individual encoder models or using weights learned by regression. We present the results of these methods against the single-encoder baselines across six diverse applications that have been studied in the literature, including predictions of house prices, poverty levels, breast cancer, pneumonia, agricultural blights, and age.

Our findings make four main contributions:

- Relying on a single encoder is risky. We find that the best-performing encoder differs across tasks. This indicates that selecting the same single encoder for different applications risks underperformance.

- Both model averaging and feature union approaches improve prediction accuracy. In each application that was tested, using multiple encoders leads to better predictive accuracy compared to using a single encoder.

- Model averaging reduces error rates across all classification tasks. Using regression to assign the weights for averaging consistently led to the best performance.

- Model performance depends on the variety of visual information contained in images, those with more variation benefit more from a feature union approach.

Together, these results indicate that using multiple encoders—either through feature union or model averaging—eliminates the need to select a single encoder, which introduces model risk. Model averaging provides a simple way to improve accuracy and robustness without requiring researchers to engage in model selection.

## 1.1 THE USE OF DEEP LEARNING MODELS IN PREDICTION APPLICATIONS

Advances in computer vision have led to growing numbers of researchers using CNNs that have been pre-trained on large databases like ImageNet (Deng et al., 2009) to support research involving classification and regression. These tools are now commonplace in research in medicine (e.g Ker et al., 2017; Sharma et al., 2023; Sajol et al., 2024; Ikechukwu et al., 2021; Al-Haija & Adebanjo, 2020), real estate (e.g Glaeser et al., 2018; Law et al., 2019; Peterson & Flanagan, 2009; Ahmed & Moustafa, 2016; Nordstrom et al., 2024), natural sciences (e.g Fu et al., 2022; Chen et al., 2021; Mukti & Biswas, 2019; Picon et al., 2022; Shah et al., 2023), and the social sciences (e.g Ludwig & Mullainathan, 2024; Compiani et al., 2023; Theckedath & Sedamkar, 2020; Yeh et al., 2020; Rabbi, 2021). In this section, we provide a brief overview of how encoders are used for image processing and how these have been implemented across fields.

### ENCODERS FOR IMAGE PROCESSING

An encoder is a neural network that transforms an input image into a set of features (an "encoding"). This is achieved by passing the input through several layers of the network, where each layer captures and represents different aspects of the data. Each encoder has a different architectural structure. This leads to different feature representations, making some encoders more suitable for specific types of tasks. To examine the best off-the-shelf tools, we therefore consider the encodings from four contemporary classifier encoders: ResNet50 (He et al., 2016), VGG16 (Simonyan & Zisserman, 2014), Inception Network (Szegedy et al., 2015), and MobileNet (Howard et al., 2017).

ResNet50 was a significant innovation in CNN encoder design, introducing residual connections that allow gradients to flow through the network directly by skipping one or more layers. This allowed networks to be deeper than was previously feasible without a corresponding decrease in performance. This has also become the default for many researchers looking for an off-the-shelf encoder to process images (Ludwig & Mullainathan, 2024; Sharma et al., 2023; Mukti & Biswas, 2019). Like ResNet50, VGG16, Inception Network (also known as GoogLeNet), and MobileNet, each uses their own novel architecture to encode an input image, resulting in different encodings. We used versions of these encoders that had been pre-trained on the ImageNet dataset (Deng et al., 2009).

In addition to encoding the entire photo using these pre-trained encoders, we also encode the individual features in each image using two panoptic segmentation encoders. Panoptic feature segmentation is a progressive methodology in computer vision that combines semantic and instance segmentation to categorize every pixel in an image into a class (e.g., "tree", "car", "building"), and distinguish individual instances of objects within the same class.

We made use of two pre-trained panoptic encoders: COCO (Common Objects in Context) (Lin et al., 2014) and Ade20K (Zhou et al., 2019). The COCO encoder is trained on an extensive range

of images featuring complex everyday scenes and provides a rich set of contexts for encoding the features in our property photos. Similarly, the Ade20K is also pre-trained on a large number of images spanning diverse settings and object classes. It is part of the MIT Scene Parsing Benchmark and is used extensively in computer vision and panoptic encoding tasks (e.g. Shao & Chung, 2024; Kumar & Raubal, 2021).

## 2 APPLICATIONS

We evaluate our models across a range of supervised learning tasks, involving six diverse applications. These applications include binary and multiclass classifications, as well as continuous regressions. These datasets span real-world domains such as medical imaging, agricultural diagnostics, satellite-based poverty estimation, and property valuation. Each dataset was processed using a standardized approach: each image was resized to $224 \times 224$ pixels, and each dataset was partitioned into training, validation, and test sets using a 70/15/15 split ratio. Table 1 summarizes the data used for each of the applications in this study.

Table 1: Application Dataset Attributes

| Dataset | Task Type | Target Type | Train/Val/Test | Total Samples |
|---|---|---|---|---|
| Breast Cancer (BreakHis) | Binary Classification | Benign/Malignant | 5536 / 1186 / 1187 | 7909 |
| Pneumonia (Chest X-ray) | Binary Classification | Normal/Pneumonia | 4099 / 878 / 879 | 5856 |
| Face Age Dataset | Multiclass Classification | 92 Age Classes | 6828 / 1464 / 1472 | 9778 |
| Rice Leaf Disease | Multiclass Classification | 4 Disease Types | 4153 / 890 / 889 | 5932 |
| WILDS-Poverty Dataset | Regression | Wealth Index (continuous) | 13768 / 2950 / 2951 | 19669 |
| House Price (Toronto) | Regression | House Price (continuous) | 5092 / 1085 / 1096 | 7273 |

### A. Breast Cancer Histopathological Dataset (BreakHis)

We employ the *BreakHis* dataset, a publicly available collection of 7,909 RGB histopathological images of breast tumor tissues obtained from 82 patients across four magnification levels (Al-Haija & Adebanjo, 2020). The dataset comprises eight distinct tumor subtypes: four benign types, and four malignant types. These subtypes reflect clinically relevant morphological variations with implications for diagnosis and treatment. To formulate a binary classification task, we converted these categories into two primary label categories: benign and malignant.

### B. Pediatric Chest X-Ray Dataset (Pneumonia)

This dataset comprises 5,856 anterior-posterior chest X-ray images of pediatric patients, labeled as either *Pneumonia* or *Normal*. The images, originally distributed across separate training, validation, and test folders, were consolidated and then randomly stratified into new splits using the 70/15/15 ratio for consistency across datasets. The dataset was sourced from Guangzhou Women and Children's Medical Center and underwent physician-based quality control (Ikechukwu et al., 2021).

### C. Face-Age Multiclass Classification Dataset

This dataset comprises 9,778 facial images annotated with chronological age ranging from 1 to 101 years (Rabbi, 2021). To mitigate the impact of sparsely represented age groups and enhance class balance, we excluded age classes with fewer than five observations, yielding 92 valid categories for our multiclass classification task.

### D. Rice Leaf Disease Image Dataset

We use 5,932 images of rice leaves categorized into four classes: *Bacterialblight*, *Blast*, *Brownspot*, and *Tungro* (Shah et al., 2023). For processing, we applied class-stratified splitting for the training, validation, and test splits.

### E. Satellite Imagery for Poverty Estimation (WILDS-Poverty)

We use the 19,669 images from the WILDS-PovertyMap dataset for predicting a continuous asset-based wealth index from RGB satellite images, originally introduced by Yeh et al. (2020). Each image represents a geographic cluster from 23 African countries. These images were extracted from

an original 8-channel Landsat product and aligned with metadata from the Demographic and Health Surveys (DHS).

**F. House Price Dataset**

This dataset includes 7,678 images of detached homes in Toronto listed between December 2018 and February 2020. The images were scraped from a publicly-accessible sales database and matched to property price data from the Multiple Listing Service (MLS) administered by the Toronto Regional Real Estate Board (Nordstrom et al., 2024) to predict house prices.

## 3 METHODOLOGY

The pipeline used to predict outcomes across these diverse applications involves three main steps. First, the images are prepared using multiple encoders. Second, the encoded features are used to predict the relevant continuous or categorical outcome relevant to each application. This can involve one of four alternative models: the use of features from a single encoder, the features from all encoders, a simple average of the predictions from the single-encoder models, or a weighted average of the predictions from the single-encoder models. Finally, the performance of these models is compared. Unlike previous research that uses deep learning to process images for predictive tasks, the central design choice of this methodology involves leveraging the features of multiple encoders simultaneously.

STEP 1: PREPARATION OF DATA USING MULTIPLE ENCODERS

Let $\mathcal{D} = \{(\mathbf{x}_i, y_i)\}_{i=1}^N$ denote a dataset where for each image $\mathbf{x_i}$, where $\mathbf{x}_i \in \mathbb{R}^{H \times W \times C}$ is an input image (RGB or multispectral) and $y_i$ is the corresponding target label.

Depending on the task, $y_i$ is either:

- A class label $y_i \in \{1, \ldots, K\}$ (for classification)
- A continuous scalar $y_i \in \mathbb{R}$ (for regression)

These extracted features are used as inputs to predict $y_i$. We extract these features using the six pre-trained backbones described in Section 1.1: ResNet50 (He et al., 2016), VGG16 (Simonyan & Zisserman, 2014), Inception Network (Szegedy et al., 2015), and MobileNet (Howard et al., 2017), and panoptic encoders COCO (Common Objects in Context) [1] (Lin et al., 2014) and Ade20K (Zhou et al., 2019). Let $\phi_m(\cdot)$ denote encoder $m \in \{1, \ldots, 6\}$, producing:

$$z_i^{(m)} = \phi_m(\mathbf{x}_i) \in \mathbb{R}^{d_m}. \tag{1}$$

The encoder extracts features $z_i{}^m = \phi(\mathbf{x}_i)$, and the task-specific head $\psi(\cdot; \theta_{\text{head}})$ then maps the encoded features $z_i^{(m)}$ to predictions $\hat{y}_i$:

$$\hat{y}_i = \psi(z_i^{(m)}; \theta_{\text{head}}), \tag{2}$$

where $\theta_{\text{head}}$ are trainable parameters. The rest of the encoder is frozen. For the Ade20K encoder, we obtain a normalized 150-bin semantic-histogram feature, which we treat as $z_i^{(m)}$, providing a compact and fixed-length summary of scene content that integrates cleanly with other encoders. Unlike previous applications, we allow $z_i{}^m$ to include the features from all encoders for an individual image, $i$, rather than the features from a single encoder.

Each of the six backbones used for encoding image features may capture different information. Most papers that have worked with images have considered only a single encoder. Others who have considered multiple encoders have regarded them as substitutes (e.g. Compiani et al., 2023). There is often a comparison of using *either* ResNet *or* Inception Network, for instance, in Gaviria Rojas et al. (2022); Shah et al. (2023); Fu et al. (2022). We take a different approach and ask whether we can do better by using the features from all encoders.

---

[1]Here, MobileNetV2 is the frozen backbone; we add a 91-unit Dense bottleneck (aligned to COCO categories) to inject lightweight object semantics that complement Ade20K scene cues and other encoders.

STEP 2: USING ENCODED FEATURES IN FOUR DISTINCT WAYS

Given the encoded features, we train four alternatives for each application. A visual representation of these different strategies can be seen in Figure 3:

**(1) Single Encoder** A head $\psi_m$ is trained on $z^{(m)}$ only:
$$\hat{y}_i^{(m)} = \psi_m\big(z_i^{(m)}\big). \tag{3}$$

**(2) Feature Union (FU, "all encoders")** Concatenate each feature from all six encoders $z_i = [z_i^{(1)}; \dots; z_i^{(6)}]$ and train a single head $\psi_{\text{FU}}$:
$$\hat{y}_i^{\text{FU}} = \psi_{\text{FU}}(z_i). \tag{4}$$

**(3) Simple Average** Average the predictions based on the six independently trained single encoder models, with equal weights:
$$\hat{y}_i^{\text{AVG}} = \frac{1}{M} \sum_{m=1}^{M} \hat{y}_i^{(m)}, \qquad M = 6 \tag{5}$$

where, for classification, the average is taken over class probabilities before argmax, and for regression it is the arithmetic mean of the scalar predictions.

**(4) Weighted Average** We also consider a weighted average prediction of the single encoder predictions:
$$\hat{y}_i^{\text{WA}} = \rho\Big([\hat{y}_i^{(1)}, \dots, \hat{y}_i^{(6)}]\Big). \tag{6}$$

For continuous outcomes, this is a linear combination over base predictions,
$$y_i = \alpha + \sum_{m=1}^{M} \beta_m \, \hat{y}_i^{(m)} + \varepsilon_i, \tag{7}$$

with the weights chosen by OLS.

For classification outcomes, we combine base probabilities via multinomial logistic regression,
$$P(y_i = k \mid \mathbf{x}_i) = \frac{\exp\Big(\alpha_k + \sum_{m=1}^{M} \beta_{mk} \, \hat{p}_{ik}^{(m)}\Big)}{\sum_{j=1}^{K} \exp\Big(\alpha_j + \sum_{m=1}^{M} \beta_{mj} \, \hat{p}_{ij}^{(m)}\Big)}. \qquad k = 1, \dots, K, \tag{8}$$

STEP 3: COMPARING MODEL PERFORMANCE

We adopt three measures to assess each model's informativeness and predictive accuracy. First, we measure the relationship between the predicted and observed outcome, $y_i$.

For regression tasks, we report (i) a per-model/encoder calibration slope obtained from an OLS regression of $y_i$ on each model's prediction,
$$y_i = \alpha_m + \beta_m \, \hat{y}_i^{(m)} + \varepsilon_i, \tag{9}$$
estimated with HC3-robust standard errors (MacKinnon & White, 1985).

For classification tasks, we report (i) a per-model/encoder class-ID calibration obtained by OLS using the true integer label on each model's predicted class ID (from softmax probabilities),
$$y_i = \alpha_m + \beta_m \, \hat{k}_i^{(m)} + \varepsilon_i, \qquad \text{where} \qquad \hat{k}_i^{(m)} = \arg\max_k \hat{p}_{ik}^{(m)}, \tag{10}$$

similarly estimated with HC3-robust standard errors, and interpreted as a coarse monotonicity check for nominal classes.

Second, we measure the $R^2$ associated with each of these regressions (depending on the task), which provides insights into how much variation the predictions from each model can explain. The final measure of model performance involves measuring the out-of-sample predictive accuracy. For regression tasks, we compute the out-of-sample mean squared error, and for classification tasks, we compute the out-of-sample error rate. The Python log for this workflow can be found in the Supplementary Material for the *Breast Cancer* application.

## 4 RESULTS

As detailed in Section 3, we evaluate six off-the-shelf visual encoders—ResNet50, VGG16, InceptionV3, MobileNet, COCO, and Ade20K—together with three feature union models: a stacked *all encoders* model, a simple model average, and a weighted average. Our dataset choices mirror widely used benchmarks and applied studies: breast cancer and pediatric chest X-rays for pathology (Al-Haija & Adebanjo, 2020; Ikechukwu et al., 2021), rice-leaf disease for agricultural diagnostics (Shah et al., 2023), satellite poverty mapping in the spirit of Yeh et al. (2020) and its WILDS packaging (Koh et al., 2020), and housing "curb appeal" for price prediction (Nordstrom et al., 2024). Residual networks remain a strong baseline across vision tasks (cf. Sarwinda et al., 2021), while distribution shift is a central concern for real-world deployment (Koh et al., 2020). We present the modest performance results in Figure 1. As an overview, the weighted average is always one of the best-performing models, though individual models or all encoders sometimes are the best.

### 4.1 PREDICTIVE PERFORMANCE (REGRESSION TASKS)

Figures 1a, 1c, and 1e report calibration, fit, and error for *Poverty Level* and *House Price*. In the *Poverty Level* task, we predict the Demographic and Health Surveys (DHS) *asset-based wealth index* at the survey-cluster level from satellite imagery. Following Yeh et al. (2020) and the WILDS PovertyMap benchmark (Koh et al., 2020), we use the benchmark-provided pooled wealth index (a standardized continuous score) and evaluate on held-out clusters. The target is interpreted as a relative measure of household wealth within and across countries, not an absolute income level.

Prior work documents that several socioeconomic indicators are visible from space: roof material (e.g., thatch vs. metal), the condition of nearby croplands, paved-road access, and connectivity to urban areas are all plausible proxies for wealth (Abelson et al., 2014). Moreover, poverty measures are spatially autocorrelated—nearby villages tend to have similar levels—so methods that exploit location metadata or post-estimation spatial smoothing can improve predictions (Jean et al., 2018; Rolf et al., 2020).

Consistent with the literature, the *Poverty Level* task is low signal in absolute fit but informative for ranked targeting. Our *weighted average* achieves the best overall performance (MAE 0.67355, $R^2 = 0.03198$) with near-perfect calibration ($\beta = 1.00000^{***}$), while the simple model average is close (MAE 0.6863, $R^2 = 0.03054$). This mirrors WILDS' findings.

Among single backbones, scene/segmentation pretraining is comparatively strong (e.g., Ade20K MSE 0.6886), supporting the role of land-cover and settlement structure in this application. Although the $R^2$ values are modest—as expected for cross-country, image-only prediction from extracted channels (e.g., RGB from unlabeled, 8-channel satellite images stored in NumPy arrays)—calibrated continuous scores are valuable as covariates or priority indices in downstream econometric or policy models.

For the *House Price dataset*, single encoders yield modest $R^2$ (e.g., Ade20K $R^2 = 0.15012$, MAE 0.3383), whereas the *all encoders* ensemble improves both fit ($R^2 = 0.24146$) and error (MAE 0.3998; MSE 0.2468) with lower calibration ($\beta = 0.5027^{***}$).

The *weighted average* is also very good with a slightly lower $R^2 = 0.23741$, but also lower error rates (MAE 0.3215, MSE 0.1699). These results align with evidence that image-based "curb appeal" can predict house prices (Nordstrom et al., 2024). Our calibration results underscore that image predictions can be improved by using multiple encoders.

Across continuous outcomes, a simple heterogeneous stack or ensemble yields both lower test error and near-unit calibration ($\beta \approx 1$). This allows the image-based score to be used directly—either as a regressor or as an operational decision score—without ad-hoc rescaling, while minimizing engineering overhead relative to training new architectures. The regression results, from which we calculate $\hat{\beta}$, and the full list of values are found in the Appendix Table 5.

### 4.2 PREDICTIVE PERFORMANCE (CLASSIFICATION TASKS)

Figures 1b, 1d, and 1f summarize calibration ($\beta$), error rates, and fit ($R^2$) on four classification benchmarks. Table 4 reports these values by dataset and model. Three facts stand out. First,

the weighted average is top or tied-top on every dataset, with the simple model *average* close behind. Second, the all encoders approach works quite poorly. Third, the strongest single encoder varies by domain. MobileNet (ImageNet) and COCO are best on both pathology datasets, while COCO (scene/segmentation pretraining) edges others on rice-leaf disease and is competitive on Age; Ade20K underperforms on medical images. Overall, ImageNet CNNs excel on texture-rich imagery, but segmentation/scene pretraining can match or exceed them when class cues include layout/context–consistent with prior reports on transfer to medical imaging and agricultural tasks (cf. Al-Haija & Adebanjo, 2020; Ikechukwu et al., 2021; Shah et al., 2023).

On *Breast Cancer*, the *weighted average* attains the lowest error (0.0868) and highest $R^2$ (0.63439), improving materially over the best single encoders (COCO: error 0.1331; $R^2 = 0.46581$, mobilenet error 0.1348, $R^2 = 0.47296$). On *Pneumonia*, weighted averaging again leads (error 0.0262; $R^2 = 0.87108$), with MobileNet and COCO close but inferior (errors 0.0364 and 0.0398). Calibration slopes are strong across models ($\beta \geq 0.9254$ for top baselines; *weighted average* $\beta = 0.9422$***). These patterns echo medical-imaging transfer results where ImageNet features paired with light heads perform strongly (Al-Haija & Adebanjo, 2020; Ikechukwu et al., 2021).

For *Rice Disease*, several encoders achieve near-ceiling performance; the *weighted average* reaches error 0.00000 and $R^2 = 1.00000$ with $\beta \approx 1$, while COCO and InceptionV3 also approach perfection (errors 0.0022–0.0045). This result aligns with leaf-disease studies reporting 98–100% accuracy for modern CNNs, particularly inceptionV3 and VGG16, on controlled imagery (Shah et al., 2023).

Treating age as 92 classes stresses both discrimination and accuracy. The *weighted average* attains the lowest error among our models (0.7595) and nearly the highest $R^2$ (0.70326), outperforming strong single encoders like COCO ($R^2 = 0.70344$, error 0.8220) and InceptionV3 ($R^2 = 0.70067$, error 0.8159). That scene/pose cues (COCO/Ade20K) help alongside texture cues (ResNet/VGG/Inception), fits the intuition that coarse geometry and context aid age grouping. Across classification, encoder choice is a high-variance decision. Simple heterogeneous aggregation substantially reduces that risk while improving accuracy and calibration. This complements the common single-encoder practice and supports treating encoder selection as a model-averaging problem rather than a one-shot choice.

Table 2: Correlation Matrices for Datasets

| | y_true | resnet50 | vgg16 | mobilenet | inception | coco | ade20k | weighted_avg | simple_avg | all_encoder |
|---|---|---|---|---|---|---|---|---|---|---|
| **Panel A: House Prices** | | | | | | | | | | |
| y_true | 1.0000 | | | | | | | | | |
| resnet50 | 0.1813 | 1.0000 | | | | | | | | |
| vgg16 | 0.0331 | -0.0083 | 1.0000 | | | | | | | |
| mobilenet | 0.2361 | 0.1199 | 0.1420 | 1.0000 | | | | | | |
| inception | 0.1509 | 0.0343 | 0.1429 | 0.2040 | 1.0000 | | | | | |
| coco | 0.2520 | 0.0824 | 0.0134 | 0.1108 | 0.1118 | 1.0000 | | | | |
| ade20k | 0.3875 | 0.2538 | -0.0660 | 0.0673 | 0.0738 | 0.1169 | 1.0000 | | | |
| weighted_avg | 0.4872 | 0.3721 | 0.0679 | 0.4845 | 0.3097 | 0.5172 | 0.7952 | 1.0000 | | |
| simple_avg | 0.3326 | 0.2471 | 0.5262 | 0.6184 | 0.6428 | 0.5095 | 0.2012 | 0.6827 | 1.0000 | |
| all_encoder | 0.4914 | 0.3273 | 0.1501 | 0.5004 | 0.4092 | 0.3839 | 0.2500 | 0.5910 | 0.6610 | 1.0000 |
| **Panel B: Pneumonia** | | | | | | | | | | |
| y_true | 1.0000 | | | | | | | | | |
| resnet50 | 0.5234 | 1.0000 | | | | | | | | |
| vgg16 | 0.8175 | 0.6316 | 1.0000 | | | | | | | |
| mobilenet | 0.9070 | 0.5150 | 0.8252 | 1.0000 | | | | | | |
| inception | 0.8596 | 0.5170 | 0.8307 | 0.8736 | 1.0000 | | | | | |
| coco | 0.8980 | 0.4877 | 0.8182 | 0.9072 | 0.8604 | 1.0000 | | | | |
| ade20k | 0.5108 | 0.2647 | 0.4187 | 0.4812 | 0.4835 | 0.4724 | 1.0000 | | | |
| weighted_avg | 0.9333 | 0.5435 | 0.8597 | 0.9558 | 0.9083 | 0.9349 | 0.5127 | 1.0000 | | |
| simple_avg | 0.9187 | 0.5515 | 0.8652 | 0.9344 | 0.9265 | 0.9239 | 0.5107 | 0.9677 | 1.0000 | |
| all_encoder | 0.2680 | 0.1926 | 0.2524 | 0.2612 | 0.2598 | 0.2182 | 0.3974 | 0.2670 | 0.2708 | 1.0000 |

## 4.3 EXPLAINING RESULTS

These results are encouraging, especially that the weighted average model is routinely either the best model, or among the best models. However, it is somewhat peculiar that the all encoders approach is among the best models for the house price dataset, and the worst for the other datasets. To explore why, consider the correlations found in Table 2. These show the correlations of the $\hat{y}_i$ values from

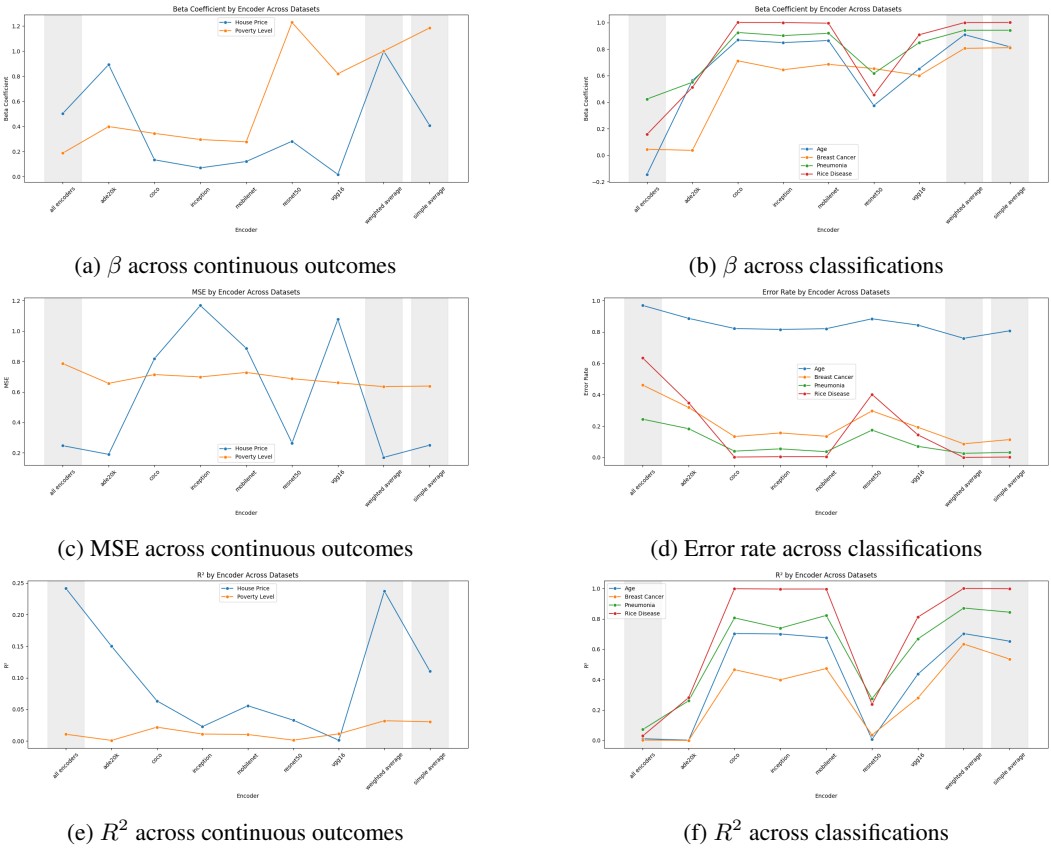

(a) $\beta$ across continuous outcomes

(b) $\beta$ across classifications

(c) MSE across continuous outcomes

(d) Error rate across classifications

(e) $R^2$ across continuous outcomes

(f) $R^2$ across classifications

Figure 1: Comparison of performance metrics across continuous outcome datasets and classification datasets.

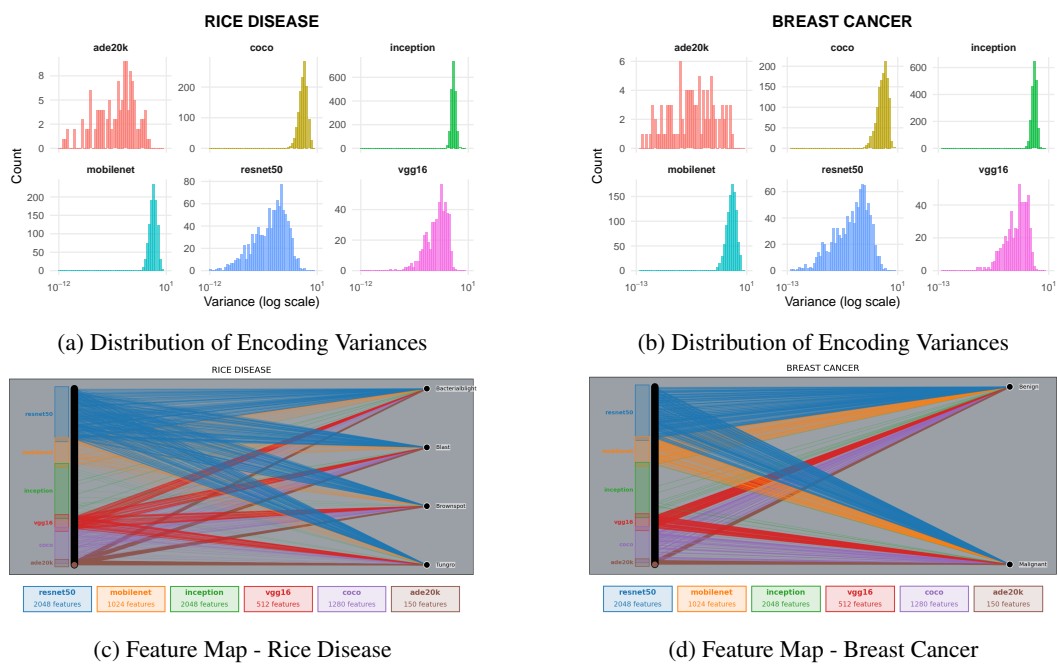

(a) Distribution of Encoding Variances

(b) Distribution of Encoding Variances

(c) Feature Map - Rice Disease

(d) Feature Map - Breast Cancer

Figure 2: Variation in Encodings and Weights Across Datasets

the different models with each and the true values $y_i$. Similar correlations for all the datasets are found in Table 3. Panel A shows the correlations for House Prices. In this, we can see that even the best models have a relatively low correlation with $y$, but both the weighted model and the all encoder model have superior correlations. Conversely, in the correlations for Pneumonia in Panel B show that the best models have very high correlations and the worst ones are noticeably lower. In this case, the weighted average improves on the best single model, and the all encoders model has a lower correlation than any single model.

To investigate why the all encoders model might fail in these settings, where some single encoder models are good, and others are bad, we plot the histogram of the variances of the encodings (across images) for both rice disease and breast cancer. These are shown in Figures 2a and 2b. In both cases, we see that the histograms are dispersed for Ade20k and ResNet50, and not very dispersed for COCO, Inception, and MobileNet. These encoders are additionally the best performing single encoder models for the classification tasks, as shown in Figure 1.

We also plot the weights of the features in the output layer of all encoding models shown in Figures 2c and 2d. The model weights suggest why all encoders can fail. There is clearly a lot of weight applied to the features from ResNet50. In both these datasets, these features have high variance across images, and the single encoder model based on these features performs poorly. We suspect that the relative strength of ResNet50 and all encoders for House Prices comes from the relative dissimilarity of images in this dataset compared to others. The house images are all of the exterior of a house, but some of these contain a car, others trees, etc. In contrast, the other datasets contain just a picture of a face or a single X-ray. The relationship between image similarity and model performance is left for future investigation.

## 5 CONCLUDING REMARKS ON ENCODER CHOICE

We show that relying on a single image encoder is risky and propose that encoder choice be treated as an ensemble problem. Prediction accuracy varies widely across encoders, and the best model differs by application. Using features from all encoders consistently reduces errors, and eliminates the need for model selection. In all applications, using model averaging led to further improvements in prediction accuracy—especially when the weights are learned by regression. These results suggest that researchers should adopt ensemble-based approaches to improve the performance of prediction tasks involving images.

## 6 STATEMENT ON USE OF LARGE LANGUAGE MODELS

We used ChatGPT to support the formatting of TikZ figures used in the manuscript. We also used ChatGPT to assist with Python coding during the analysis. The authors reviewed and edited the code and TikZ commands carefully and take full responsibility for the content of the published article.

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

# A APPENDIX

## A.1 ADDITIONAL CORRELATION TABLES

### Table 3: Correlation Matrices for Additional Datasets

| | | | | | **Panel A: Poverty** | | | | | |
|---|---|---|---|---|---|---|---|---|---|---|
| | y_true | resnet50 | vgg16 | mobilenet | inception | coco | ade20k | weighted_avg | simple_avg | all_encoder |
| y_true | 1.0000 | | | | | | | | | |
| resnet50 | 0.0357 | 1.0000 | | | | | | | | |
| vgg16 | 0.1061 | 0.2885 | 1.0000 | | | | | | | |
| mobilenet | 0.1011 | 0.1699 | 0.3253 | 1.0000 | | | | | | |
| inception | 0.1048 | 0.0915 | 0.2048 | 0.1743 | 1.0000 | | | | | |
| coco | 0.1478 | 0.1932 | 0.3468 | 0.3033 | 0.1853 | 1.0000 | | | | |
| ade20k | 0.0296 | 0.0223 | 0.1189 | 0.1110 | 0.0558 | 0.1037 | 1.0000 | | | |
| weighted_avg | 0.1788 | 0.1998 | 0.5934 | 0.5653 | 0.5862 | 0.8264 | 0.1656 | 1.0000 | | |
| simple_avg | 0.1748 | 0.2776 | 0.5506 | 0.6924 | 0.6013 | 0.7508 | 0.2207 | 0.9772 | 1.0000 | |
| all_encoder | 0.1030 | 0.1187 | 0.1613 | 0.4661 | 0.4284 | 0.3847 | 0.0861 | 0.5461 | 0.5859 | 1.0000 |
| | | | | | **Panel B: Age** | | | | | |
| | y_true_test | resnet50 | vgg16 | mobilenet | inception | coco | ade20k | weighted_avg | simple_avg | all_encoder |
| y_true_test | 1.0000 | | | | | | | | | |
| resnet50 | 0.0843 | 1.0000 | | | | | | | | |
| vgg16 | 0.6604 | 0.0855 | 1.0000 | | | | | | | |
| mobilenet | 0.8222 | 0.0797 | 0.6550 | 1.0000 | | | | | | |
| inception | 0.8371 | 0.0972 | 0.6603 | 0.8059 | 1.0000 | | | | | |
| coco | 0.8387 | 0.0915 | 0.6792 | 0.7958 | 0.7978 | 1.0000 | | | | |
| ade20k | 0.0440 | -0.0058 | -0.0185 | 0.0497 | 0.0261 | 0.0276 | 1.0000 | | | |
| weighted_avg | 0.8386 | 0.1137 | 0.7008 | 0.8553 | 0.8468 | 0.8179 | 0.0410 | 1.0000 | | |
| simple_avg | 0.9559 | 0.1632 | 0.4722 | 0.7242 | 0.7025 | 0.6760 | . | 0.7993 | 1.0000 | |
| all_encoder | -0.1060 | -0.0154 | -0.1331 | -0.1281 | -0.1372 | -0.1129 | 0.0255 | -0.1150 | -0.1839 | 1.0000 |
| | | | | | **Panel C: Breast Cancer** | | | | | |
| | y_true_test | resnet50 | vgg16 | mobilenet | inception | coco | ade20k | weighted_avg | simple_avg | all_encoder |
| y_true_test | 1.0000 | | | | | | | | | |
| resnet50 | 0.1899 | 1.0000 | | | | | | | | |
| vgg16 | 0.5285 | 0.1435 | 1.0000 | | | | | | | |
| mobilenet | 0.6877 | 0.1750 | 0.5099 | 1.0000 | | | | | | |
| inception | 0.6319 | 0.1726 | 0.4766 | 0.5617 | 1.0000 | | | | | |
| coco | 0.6825 | 0.1971 | 0.4888 | 0.6028 | 0.5494 | 1.0000 | | | | |
| ade20k | 0.0103 | 0.0305 | -0.0355 | -0.0047 | -0.0122 | 0.0085 | 1.0000 | | | |
| weighted_avg | 0.7965 | 0.1823 | 0.5984 | 0.7578 | 0.7215 | 0.7816 | 0.0283 | 1.0000 | | |
| simple_avg | 0.7318 | 0.2091 | 0.6186 | 0.7042 | 0.6830 | 0.7696 | 0.0073 | 0.8254 | 1.0000 | |
| all_encoder | 0.0491 | -0.0254 | 0.0625 | 0.0168 | 0.0366 | 0.0237 | 0.0511 | 0.0601 | 0.0461 | 1.0000 |
| | | | | | **Panel D: Rice Disease** | | | | | |
| | y_true_test | resnet50 | vgg16 | mobilenet | inception | coco | ade20k | weighted_avg | simple_avg | all_encoder |
| y_true_test | 1.0000 | | | | | | | | | |
| resnet50 | 0.4882 | 1.0000 | | | | | | | | |
| vgg16 | 0.9010 | 0.5282 | 1.0000 | | | | | | | |
| mobilenet | 0.9982 | 0.4910 | 0.9004 | 1.0000 | | | | | | |
| inception | 0.9982 | 0.4839 | 0.9027 | 0.9963 | 1.0000 | | | | | |
| coco | 0.9991 | 0.4866 | 0.9017 | 0.9973 | 0.9991 | 1.0000 | | | | |
| ade20k | 0.5309 | 0.2617 | 0.4839 | 0.5300 | 0.5301 | 0.5295 | 1.0000 | | | |
| weighted_avg | 1.0000 | 0.4882 | 0.9010 | 0.9982 | 0.9982 | 0.9991 | 0.5309 | 1.0000 | | |
| simple_avg | 0.9991 | 0.4866 | 0.9017 | 0.9973 | 0.9991 | 1.0000 | 0.5295 | 0.9991 | 1.0000 | |
| all_encoder | 0.1731 | 0.0069 | 0.1488 | 0.1738 | 0.1725 | 0.1713 | 0.1495 | 0.1731 | 0.1713 | 1.0000 |

## A.2 REGRESSION TABLES

Table 4: Classification Performance by Encoder

| | resnet50 | mobilenet | inception | vgg16 | coco | ade20k | all encoders | NN avg | Simple avg |
|---|---|---|---|---|---|---|---|---|---|
| **Panel A: Age ($N$= 9,778)** | | | | | | | | | |
| Beta | 0.3750*** | 0.8649*** | 0.8489*** | 0.6517*** | 0.8686*** | 0.5629 | -0.1440*** | 0.9090*** | 0.8171*** |
| Error Rate | 0.8838 | 0.8207 | 0.8159 | 0.8438 | 0.8220 | 0.8865 | 0.9688 | 0.7595 | 0.8071 |
| MSE | 1353.91 | 212.16 | 196.54 | 477.26 | 190.96 | 1411.31 | 1182.16 | 212.50 | 243.46 |
| $R^2$ | 0.00711 | 0.67602 | 0.70067 | 0.43615 | 0.70344 | 0.00193 | 0.01124 | 0.70326 | 0.65320 |
| **Panel B: Breast Cancer ($N$= 7,909)** | | | | | | | | | |
| Beta | 0.6533*** | 0.6857*** | 0.6445*** | 0.6012*** | 0.7121*** | 0.0372 | 0.0457* | 0.8057*** | 0.8109*** |
| Error Rate | 0.2965 | 0.1348 | 0.1559 | 0.1921 | 0.1331 | 0.3184 | 0.4608 | 0.0868 | 0.1137 |
| MSE | 0.2965 | 0.1348 | 0.1559 | 0.1921 | 0.1331 | 0.3184 | 0.4608 | 0.0868 | 0.1137 |
| $R^2$ | 0.03608 | 0.47296 | 0.39929 | 0.27930 | 0.46581 | 0.00011 | 0.00241 | 0.63439 | 0.53557 |
| **Panel C: Pneumonia ($N$= 5,856)** | | | | | | | | | |
| Beta | 0.6166*** | 0.9195*** | 0.9017*** | 0.8491*** | 0.9254*** | 0.5504*** | 0.4238*** | 0.9422*** | 0.9424*** |
| Error Rate | 0.1741 | 0.0364 | 0.0546 | 0.0705 | 0.0398 | 0.1832 | 0.2435 | 0.0262 | 0.0319 |
| MSE | 0.1741 | 0.0364 | 0.0546 | 0.0705 | 0.0398 | 0.1832 | 0.2435 | 0.0262 | 0.0319 |
| $R^2$ | 0.27398 | 0.82259 | 0.73892 | 0.66830 | 0.80646 | 0.26095 | 0.07184 | 0.87108 | 0.84395 |
| **Panel D: Rice Disease ($N$= 5,932)** | | | | | | | | | |
| Beta | 0.4548*** | 0.9955*** | 1.0000*** | 0.9083*** | 1.0008*** | 0.5118*** | 0.1575*** | 1.0000*** | 1.0008*** |
| Error Rate | 0.4016 | 0.0045 | 0.0045 | 0.1440 | 0.0022 | 0.3487 | 0.6333 | 0 | 0.0022 |
| MSE | 1.3487 | 0.0045 | 0.0045 | 0.2418 | 0.0022 | 1.2148 | 2.2722 | 0 | 0.0022 |
| $R^2$ | 0.2383 | 0.99634 | 0.99632 | 0.81179 | 0.99816 | 0.2818 | 0.02995 | 1.0000 | 0.99816 |

*Notes:* Error rate $= 1 -$ accuracy. $\beta$ is the calibration slope (true on predicted). *Significance:* $^{*}\, p < 0.10$, $^{**}\, p < 0.05$, $^{***}\, p < 0.01$ for $\beta$.

Table 5: Regression Performance Across by Encoder

| | resnet50 | mobilenet | inception | vgg16 | coco | ade20k | all encoders | NN avg | Simple avg |
|---|---|---|---|---|---|---|---|---|---|
| **Panel A: House Price ($N$= 7,273)** | | | | | | | | | |
| Beta | 0.2808*** | 0.1194*** | 0.0689*** | 0.0167 | 0.1329*** | 0.8918*** | 0.5027*** | 1.0000*** | 0.4082*** |
| MAE | 0.4031 | 0.7238 | 0.8316 | 0.8226 | 0.7176 | 0.3383 | 0.3998 | 0.3215 | 0.3875 |
| MSE | 0.2648 | 0.8874 | 1.1681 | 1.0765 | 0.8195 | 0.1900 | 0.2468 | 0.1699 | 0.2501 |
| $R^2$ | 0.03286 | 0.05573 | 0.02276 | 0.00110 | 0.06350 | 0.15012 | 0.24146 | 0.23741 | 0.11064 |
| **Panel B: Poverty Level ($N$= 19,669)** | | | | | | | | | |
| Beta | 1.2278** | 0.2767*** | 0.2946*** | 0.8193*** | 0.3437*** | 0.3982 | 0.1867*** | 1.0000*** | 1.1841*** |
| MAE | 0.7331 | 0.7369 | 0.6770 | 0.6651 | 0.7173 | 0.6886 | 0.7375 | 0.6735 | 0.6863 |
| MSE | 0.6871 | 0.7282 | 0.6986 | 0.6604 | 0.7142 | 0.6565 | 0.7861 | 0.6348 | 0.6388 |
| $R^2$ | 0.00128 | 0.01022 | 0.01099 | 0.01126 | 0.02184 | 0.00088 | 0.01061 | 0.03198 | 0.03054 |

*Notes:* MAE/MSE are test-set errors (lower is better). $\beta$ is the calibration slope (true on predicted). *Significance:* $^{*}\, p < 0.10$, $^{**}\, p < 0.05$, $^{***}\, p < 0.01$ for $\beta$.

## A.3 ENCODINGS DIAGRAM

Figure 3: Image Encoding Comparisons

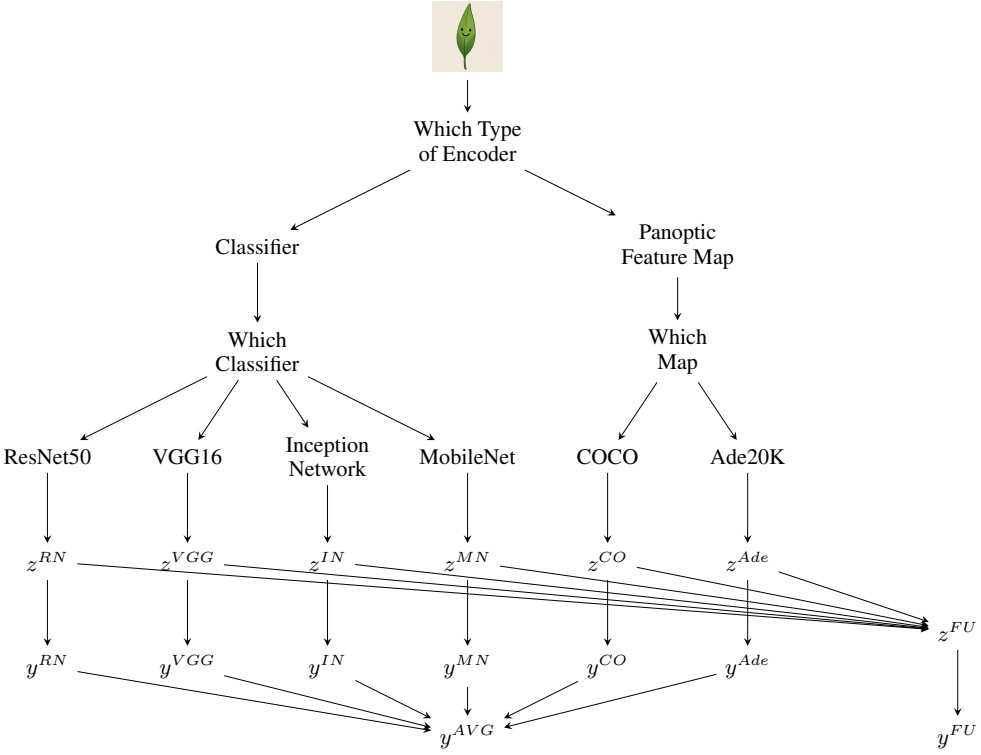

Figure 4: Note: This figure describes how an image is encoded by either a classifier or a panoptic feature map. The resulting encodings ($z^{RN}, z^{VGG}, z^{IN}, z^{MN}, z^{CO}, z^{Ade}$) produce outputs ($y^{RN}, y^{VGG}, y^{IN}, y^{MN}, y^{CO}, y^{Ade}$) that can be combined either via (simple) model averaging ($y^{AVG}$), or through a feature union representation ($y^{FU}$) of all encoders.

