# OpenReview forum: "Take Another Look: Improving Information Extraction From Images With Multiple Encoders"
_ICLR.cc/2026/Conference — Submitted to ICLR 2026_

### Official Review · Reviewer_wME9 · 2025-10-24

**Soundness:** 2
**Presentation:** 2
**Contribution:** 1
**Rating:** 2
**Confidence:** 5

**Summary:**

The paper presents an empirical study on how to combine frozen predictions from one/multiple encoders and test this application on different datasets belonging to different domains.

**Strengths:**

**S1.** The objective of the paper is relevant, since selecting the appropriate pre-trained model from a pool of models and how to optimally derive the features for downstream tasks represents a relevant research question

**S2.** The experiments are run on 6 different datasets coming from different domains, and 5 CNN models, which represent a sufficiently large experimental setup

**S3.** The paper reads fairly well, and it is not badly written

**Weaknesses:**

The paper has critical weaknesses (W), particularly **W1** and **W2**, that largely outweigh its strengths, rendering it unsuitable for publication at ICLR in its current form.

**W1.** Novelty of contributions (C): It is well known within the community that different encoders may yield suboptimal performance for various downstream tasks due to differences in pre-training data/ architecture/ training algorithm, etc. (C1).   Additionally, it is expected that model/feature averaging will lead to better performance, as averaging reduces variance (C2 and C3). Hence, ¾ contributions do not introduce relevant new insights beyond existing knowledge in the field.

**W2.** Coverage and evaluation of related work: A large body of work has studied model selection from a “model zoo” (e.g., see [1,2] and references therein). The paper is missing both a literature description of related works and an evaluation of such baselines against their methodology.

[1] LogME: Practical Assessment of Pre-trained Models for Transfer Learning, ICML 2021

[2] Exploiting Model Zoo for Out‑of‑Distribution Generalization, NeurIPS 2022

**W3.** Schematics presentation: Both Figures 1 and 2 are hard to understand without explanatory captions, difficult to read given the small fonts, and do not respect presentation standards for a conference of the level of ICLR

**Questions:**

I appreciate the authors’ effort and recognize the relevance of the topic.

However, the main concerns (limited novelty, missing literature coverage, and lack of comparison to prior feature-based model-selection methods) reflect substantive rather than clarificatory issues. Therefore, I do not have specific questions for rebuttal that would likely alter my assessment.

---

### Official Review · Reviewer_6A1Z · 2025-10-27

**Soundness:** 2
**Presentation:** 2
**Contribution:** 2
**Rating:** 2
**Confidence:** 4

**Summary:**

This paper studies the possibility of ensembling multiple pretrained image encoders to generate predictive feature sets that improve performance in image tasks. Two strategies are studied: concatenating features from different encoders ("feature union"), and averaging the predictive model outputs, either with a simple average or a weighted average where the weights in the average are learned. These methods are evaluated on 2 regression datasets and 4 classification datasets, using 6 "off the shelf" image encoders: ResNet50, VGG16, InceptionV3, MobileNet, COCO, and Ade20K. The paper claims that combining the embeddings results in better predictive models than using single embeddings alone, which seems to be generally backed up by the experimental results in the paper.

**Strengths:**

1. I see the main motivation of this paper being to help users -- probably users from domains outside of ML, given the introduction's focus on medical/social good applications -- avoid an expensive part of the ML pipeline: choosing which model to use to encode their data (here, the focus is on image encoding). Rather than propose a new image encoding method, the authors consider the possibility of ensembling multiple existing encoding methods. I think there is a **strength in the simplicity of the proposed approach**, which could be very easy for users of all types to implement.

2. Another strength is the **diversity of datasets used for assessing model performance**, spanning many different regression and classification tasks. However, this strength is dampened a bit without a connection to how other proposed models have performed on these datasets (see weaknesses below).

3. The **diversity of image encoders** -- including feature extractors and panoptic segmentation models -- is a plus. Though, these are all relatively older methods (though still reasonable choices) -- see boxes below.

4. Presentation of written text: the paper is clearly written and easy to understand.

**Weaknesses:**

There are several major weaknesses of this paper in it's current form. I list what I see to be the most major weaknesses in the hopes that these are constructive to the authors in a potential revision of the work. I do not think the authors will be able to address these major weaknesses in a rebuttal period,

**Major Weaknesses**:
1. There is no related work section! Model ensembling has a rich history in ML, and it's completely missing from this paper. A discussion of more recent related work for fusing outputs from multiple encoders is also missing.
2. The models compared are all from 2019 or earlier, limiting the scope of the results. Would these ensembling methods make a difference if the encoders themselves were achieving higher accuracy individually? It's impossible to tell from the results provided.
3. There is no comparison to past approaches on the datasets provided, and it's not clear if the train/val/test splits here are consistent with past work. For example, I believe that performance achievable for the povertymap dataset is much higher than what is reported here -- the authors could easily have compared to the performance of other methods on the leaderboard, for example, to contextualize whether their low predictive performance is expected.
4. The results could be better tied into the motivation of the paper.  The motivation -- as I see it -- is to help users avoid the need to try different model architectures. But there is no discussion of the costs associated with ensembling -- just as relying only on a resnset50 isn't the best approach, neither is assuming that ensembling a bunch of models is the right way to go. There's no discussion of (i) the added computational costs and complexity of training and combining multiple models or when ensembling is most likely to help, for example. Given that ensembling is expected to help (see: rich history of ensembling methods in ML), these more in-depth takes would be important if the goal is to contribute user-friendly approaches that are shown to be robust across datasets and tasks. I just don't think the paper has achieved that yet.

**Minor Weaknesses**
1. The figures and tables do not stand on their own - I suggest to add more detail to the captions.
2. Figure 1 should not be a line plot, the horizontal axis variables are categorical, not numeric.

**Questions:**

I have several questions as well for the authors that I hope can be addressed during the rebuttal and discussion phase:

1. In figure one panels (c) and (e), something looks off. Since R2 score is a rescaled version of MSE, I think the methods that have higher MSE should always have higher R2 score on a given dataset (blue or orange line), but this isn't always the case. Could the authors look into this and clarify?
2. Why not test more recent encoding methods, like ViTs, for example?
3. Why are the authors reporting R2 on classification tasks? Would there not be a more suitable score like F1 score or AUC?
4. Did the authors run multiple random trials for experiments? If so, the figures are missing standard errors / standard deviations that are necessary to evaluate their claims.

---

### Official Review · Reviewer_bmWn · 2025-10-31

**Soundness:** 2
**Presentation:** 1
**Contribution:** 1
**Rating:** 0
**Confidence:** 4

**Summary:**

This paper examines a few combinations of encoders for both classification and regression models across differeng tasks. Results indicate that combining various encoders might be beneficial compared with leveraging a single image encoder.

**Strengths:**

1. The authors conduct experiments on various tasks and dataset, somehow bringing observations and insights regarding the utility of image encoders.

**Weaknesses:**

1. The major focus of this research is vague. It seems that the major purpose of this study is to examine some existing models rather than propose novel method / introduce knowledge. Notably, the observations indicated in this study have already been widely noticed and accepted by the community, such as the usefulness of aggregating encoders. Therefore, this study leaves me the impression that it is a project report rather than a research paper, with marginal research contribution.

2. The presentation is not very rigor. For instance, in Table 1, only the first column (comparison between prediction and ground truth) is valuable, while the comparison between outcome of different predictions does not contribute to the evaluation of encoders. In Figure 1, the authors choose to utilize line chart rather than bar chart to present results from individual encoders, which can be somewhat misleading as their performance are strictly independent. In Eq. (8), $\hat p_{ij}^{(m)}$ has not been previously defined, which should be referred as $\hat y$ instead.

3. Moreover, organization of this paper is messy. I am assuming "THE USE OF DEEP LEARNING MODELS IN PREDICTION APPLICATIONS" and "ENCODERS FOR IMAGE PROCESSING" should serve as two subsections of the absent section of "Related Work", rather than two paragraphs in Introduction.

**Questions:**

Please refer to my aforementioned weaknesses. Any clarification will be helpful.

---

### Official Review · Reviewer_ap1e · 2025-10-31

**Soundness:** 1
**Presentation:** 2
**Contribution:** 1
**Rating:** 2
**Confidence:** 4

**Summary:**

In this paper, the authors focus on using multiple off-the-shelf pre-trained CNN and panoptic encoders for feature extraction on multiple datasets. Then they aim to use these features ensembled outperform single-encoder models across various tasks. They provide with empirical results and analysis to prove their insights.

**Strengths:**

- The paper is clearly written, with a clear tone and details.

**Weaknesses:**

- The paper mainly ensembles Resnet50, VGG16, inception, mobilenet, coco, and ade20k, without using any state-of-the-art transformer-based encoders, features, or recent hybrid CNN-transformers- which should be considered a baseline experiment.
- The paper does not demonstrate any technical innovation or explanation:: in model design or algorithm or feature concatenation method. It basically provides domain-specific empirical results for simple feature usage method from multiple CNN based encoders.
- The datasets used in the paper, is very small and domain specific supervised datasets. It lacks diversity and generalizabiity. Does the claim remain true if it is used on more recent or more used datasets?
- The paper lacks an ablation study for failure cases, if and when the method will fail.
- Using multiple frozen encoders means users have to save each of the models and do inference for one image multiple times. This is both resource and time-intensive with the gain being relatively small.
- The encoder ensembling topic has extensive prior work, including feature fusion and multi-layer aggregation techniques(cross layer attention, hierarchical ensembling). None of these has been reviewed or benchmarked in this work.
- The paper does not convey well, the domain significance, broader impact of the work.

**Questions:**

- Comparison with the CNN-transformer architecture to show if the results hold.
- ablation study, on different model sizes, model train strategies, and model type: how would this affect the performance gain?
- Usage of multiple bigger datasets for classification to show the generalizability of the claim.
- Instead of focusing on dataset explanation in the paper, in my humble opinion, more focus should be on using multiple models and an ablation study.

---

### Meta-Review · Area_Chair_jGrz · 2026-01-07

**Summary:**

This paper presents to use multiple encoders to outperform single encoders and outperform existing methods. Reviewers raised several concersn regarding the key novelty of the paper and the presentation of the results:
- The lack of experimental results / ablation study. The related state-of-the-art ensemble methods are not fully compared or discussed.
- The observation of the paper has already been studied and known.
- Missing related work section.

**Reviewer Concerns:**

No rebuttal is presented.

**Reviewer Scores:**

No change of score due to the absence of rebuttal.

---

### Decision · Program_Chairs · 2026-01-26

Reject